# The Next Frontier in Pancreatic Cancer: Targeting the Tumor Immune Milieu and Molecular Pathways

**DOI:** 10.3390/cancers14112619

**Published:** 2022-05-25

**Authors:** Chao Yin, Ali Alqahtani, Marcus S. Noel

**Affiliations:** Ruesch Center for the Cure of Gastrointestinal Cancers, Lombardi Comprehensive Cancer Center, Georgetown University, Washington, DC 20007, USA; chao.yin@gunet.georgetown.edu (C.Y.); ali.z.alqahtani@medstar.net (A.A.)

**Keywords:** pancreatic cancer, immunotherapy, targeted therapy, tumor microenvironment

## Abstract

**Simple Summary:**

Pancreatic ductal adenocarcinoma (PDAC) has a notoriously bad prognosis due to its high mortality and lack of good therapies. Chemotherapy is the current standard of treatment for PDAC, yet survival for most PDAC remain at around one year. Better therapeutic options are in dire need. Unlike other cancer types where targeted therapies and immunotherapies have changed the treatment landscape, their uses in pancreatic cancer are limited. However, there is increasing evidence in preclinical and early clinical studies that suggest these agents hold the key to the next frontier in PDAC treatment. We herein review some selected evidence.

**Abstract:**

Pancreatic ductal adenocarcinoma (PDAC) is an aggressive cancer with abysmal prognosis. It is currently the third most common cause of cancer-related mortality, despite being the 11th most common cancer. Chemotherapy is standard of care in all stages of pancreatic cancer, yet survival, particularly in the advanced stages, often remains under one year. We are turning to immunotherapies and targeted therapies in PDAC in order to directly attack the core features that make PDAC notoriously resistant to chemotherapy. While the initial studies of these agents in PDAC have generally been disappointing, we find optimism in recent preclinical and early clinical research. We find that despite the immunosuppressive effects of the PDAC tumor microenvironment, new strategies, such as combining immune checkpoint inhibitors with vaccine therapy or chemokine receptor antagonists, help elicit strong immune responses. We also expand on principles of DNA homologous recombination repair and highlight opportunities to use agents, such as PARP inhibitors, that exploit deficiencies in DNA repair pathways. Lastly, we describe advances in direct targeting of driver mutations and metabolic pathways and highlight some technological achievements such as novel KRAS inhibitors.

## 1. Introduction

Pancreatic ductal adenocarcinoma (PDAC) is the most common malignancy of the pancreas and is associated with abysmal prognosis. Around 62,210 new cases of pancreatic cancer (PDAC accounts for >90% of pancreatic cancers) were estimated in the United States in 2022, accounting for 49,830 deaths in the same year [1]. Even though it is the 11th most common cancer in the surveillance, epidemiology, and end results (SEER) database, it is currently the third most common cause of cancer-related mortality and is projected to become the second most common cancer-related mortality by 2030 [1,2,3]. Only 20% of PDAC is diagnosed at an early stage, where it potentially resectable and curable; 30% are diagnosed at a locally advanced stage and not amenable to surgery, and 50% are metastatic [4]. Even in resected cancers, systemic recurrence rates are as high as 80–90% [5]. For patients who have unresectable disease, chemotherapy is the cornerstone of management. However, despite improvements in combination chemotherapy, such as front-line modified dosing of fluorouracil, oxaliplatin, and irinotecan (mFOLFIRINOX) or gemcitabine and nab-paclitaxel (GemNab), median overall survival (mOS) for metastatic PDAC is less than one year and only slightly higher for locally advanced unresectable PDAC [6,7].

Better systemic therapy options are needed beyond traditional chemotherapy. While immunotherapies and targeted therapies are becoming mainstay treatment options in various other solid tumors, there are currently no front-line non-chemotherapy options in PDAC. The U.S. Food and Drug Administration (USFDA) has only approved pembrolizumab, larotrectinib, entrectinib, and olaparib as immunotherapy and targeted therapy options in subsequent-line settings for PDAC [8]. In this review, we discuss the evolving landscape of therapeutic targets in PDAC, including selected clinical data from corresponding trials.

## 2. Immunotherapy

### 2.1. Immune Checkpoint Inhibitors (ICIs) and the Tumor Microenvironment (TME)

Unlike immunogenic tumors, such as renal cell carcinoma and melanoma, where ICIs have made a notable positive impact on survival, PDAC has remained largely refractory to many immunotherapies [9,10]. The KEYNOTE-158 basket trial was widely regarded as a landmark trial that led to USFDA approval of pembrolizumab (anti-PD-1) across high microsatellite instability (MSI-H) advanced solid tumors [11]. There were 22 patients in the trial with PDAC, with an overall response rate (ORR) of 18.2%, which was much lower than other cohorts in the study such as in gastric or cholangiocarcinoma (ORR of 45.8% and 40.9%, respectively). Even so, only a very small proportion of real-world PDAC, ≈1–2%, are MSI-H [12]

Similarly, a combination of anti-CTLA-4 and anti-PD-1/PD-L1 approaches in phase I/II trials have failed to demonstrate the same degree of efficacy in PDAC as compared to other tumor types [13]. Trials have also showed the limited benefit in adding ICIs, such as pembrolizumab or durvalumab (anti-PD-L1), to chemotherapy backbones [14]. For example, in a two-armed phase II trial of GemNab with or without durvalumab plus tremelimumab (anti-CTLA-4) in metastatic PDAC, there was no added benefit from immunotherapy in mOS (9.8 months in immunotherapy arm vs. 8.8 months in control arm; HR 0.94, *p* = 0.72) or mPFS (5.5 vs. 5.4 months, respectively, HR 0.98, *p* = 0.91) [15]. Another phase III study from China showed that the addition of sintilimab (anti-PD-1) may improve ORR (50% vs. 23.9%, *p* = 0.10), but did not improve mOS (10.9 vs. 10.8 months, HR 1.083, 95% CI 0.68–1.69) [16]. We have made a comprehensive list of reported and ongoing ICI trials in PDAC in Table 1 and Table 2, respectively. On the basis of these results, it was heavily suggested that PDAC has immune features that are different from other types of solid tumors.

An understanding of the TME of PDAC helps to explain some of its refractoriness to immunotherapy (Figure 1). As demonstrated in both human and mouse models, a hallmark of the PDAC TME is an abundance of stroma, which encompasses the non-cancer cell components of the TME. The dense desmoplasia of the stroma includes cellular and molecular components that inhibit both spontaneously and therapeutically induced anti-tumor immunity [47]. Pancreatic stellate cells (PSCs) are a unique component of a normal pancreas and play a vital role in tumoral propagation. PSCs and activated fibroblasts derived from PCSs secrete abundant proteins that help form the extracellular matrix during tumorigenesis [48,49]. Both in vitro and in vivo, PSCs interact with a variety of immune cells, including T cells and macrophages. For example, PSCs secrete the chemokine CXCL12, which has a chemotactic effect on CD8+ T cells and may explain the frequent sequestration of CD8+ T cells observed in the stroma rather than their accumulation next to tumor cells [50,51]. Cancer stem cells (CSCs), although not unique to PDAC, are another class of cells with immune-evasive potential. Recent data have suggested the interplay of CSCs with the TME, whereby CSCs help create an immunosuppressive milieu that in turn helps to potentiate its own expansion [52]. Some of the immunosuppressive properties of CSCs include impaired antigen presentation, downregulation of tumor-associated antigens, and inhibition of cytotoxic granules [53]. Pertaining to PDAC, Kim and colleagues identified a group of PDAC cells with CSC features, namely, increased aldehyde dehydrogenase (ALDH), a marker of stem/progenitor cells. The authors found that although these cells predicted resistance to anti-tumor therapies, they may be suppressed by disulfiram [54].

Indeed, PDAC is characterized as a “cold tumor” due to its paucity of intra-tumoral CD8+ T cells in both human tumor samples and mouse models [55,56]. PDAC cells also suppress is own MHC I expression to prevent recognition by CD8+ T cells, facilitating their immune evasion [57]. These characteristics explain the decreased response to ICIs, given the therapeutic effects of ICIs are mediated by CD8+ T cells. Furthermore, regulatory T cells (Tregs) are recruited into the TME and play an immunosuppressive role by overexpression of transcription factor fork-head-box protein 3 (FOXP3). A decrease in CD8+ T cells and increase in Tregs have also been associated with poorer prognosis in PDAC [56]. In fact, Kieler et al. summarized the mechanism of immune escape in PDAC as attributed to low mutational load, impaired function of dendritic cells, CTLA-4 and PD-1/PD-L1 signaling and upregulation, trafficking of Tregs into the TME, reduced migratory ability of CD8+ T-cells due to dense stroma, and downregulation of MHC-I molecules [58]. Furthermore, the state of T cell exhaustion is relevant to PDAC, whereby T cells in chronic inflammatory states, such as in the case of cancer, become dysfunctional due to chronic antigen exposure. As such, the effector T cell function is hindered by imbalance of inhibitory and stimulatory signals, namely, an increase in multiple inhibitory receptors such as PD-1, CTLA-4, TIM-3, and LAG-3 [59]. The complex interplay of immunosuppressive cells, including Tregs, tumor-associated macrophages (TAMs), myeloid-derived suppressor cells (MDSCs), and regulatory B cells (Bregs) also contribute to T cell exhaustion, which in turn contributes to PDAC resistance to many immunotherapies that rely on healthy T cells [59]. At the same time, the improved understanding of the tumor biology in PDAC has prompted the investigation of combination strategies by incorporating novel classes of immunotherapies (i.e., vaccines or cellular therapies) with ICIs that may further activate and prime T cells in the TME.

### 2.2. Vaccine Therapies

A number of vaccine therapies have been conducted on PDAC, but so far with very limited success [10]. Therapeutic vaccines include whole-cell, dendritic cell (DC), and DNA/peptide vaccines. GVAX is an example of an irradiated allogenic whole tumor cell vaccine that is engineered to express granulocyte macrophage colony-stimulating factor (GM-CSF) in order to stimulate antigen uptake by antigen-presenting cells (APC) to promote T cell priming. This was studied in phase II trials (NCT0084383 and NCT0141700) in combination with CRS-207 (a live attenuated Listeria monocytogenes vaccine designed to stimulate immune response) and seemed to demonstrate improved OS in single-arm studies [60,61]. However, ultimately, the combination of GVAX and CRS-207 plus chemotherapy was not shown to improve survival over chemotherapy alone [62].

KRAS vaccines and the GV1001 vaccine (including fragments of hTERT protein found in large portions of PDAC cells) are examples of peptide vaccines, but have also yielded disappointing results so far in larger clinical trials [13]. A phase III study of GV1001 plus gemcitabine and capecitabine in PDAC did not improve OS compared to chemotherapy alone [63]. Despite some of the disappointments of these vaccines with respect to OS, there are still promising data from these trials suggesting that the vaccines could generate robust T-cell responses to tumor neoantigens and lead to T-cell infiltration into PDAC tumors [64]. There are ongoing studies looking at ways to improve vaccine efficacy. For example, it was noted that vaccine therapy induces upregulation of the PD-L1 pathway, and hence trials combining vaccines and PD-L1 checkpoint blockade are underway [64,65]. We have listed reported and ongoing vaccine trials in Table 3 and Table 4, respectively.

### 2.3. Cellular Therapies

Cellular therapy in solid tumors can be represented by chimeric antigen receptor T cell (CAR-T) and adoptive transfer of tumor-infiltrating lymphocytes (TILs). Initial CAR-T development used CD19 and CD20 as targets in hematologic malignancies with excellent results, but new targets are necessary for solid tumors. Some engineered CAR-T targets that showed efficacy in mouse models include CEA, mesothelin, EGFR, and HER2 [14]. Unfortunately, this efficacy has not been replicated in clinical trials [108]. Several challenges were presented. The aforementioned T-cell exhaustion could affect the quality of innate T cells that are harvested from the host, such that the ability for an ideal CAR-T to infiltrate the tumor and propagate in the TME may be hindered by exhausted adoptive T cells [59]. Furthermore, improved specificity of the chosen CAR-T target is necessary to prevent unwanted side effects. For example, targeting the ubiquitous HER2 results in autoimmunity in healthy cells, such as epithelial and skin cells [64].

Cellular therapies continue to evolve. For example, ongoing studies show some promise in adding tumor-targeting cytokine receptors to CAR-T to help with intratumoral trafficking and tumoral response [64]. Another phase I trial of adoptively transferred, autologous, nonengineered, multiantigen specific T cells demonstrated good safety and tolerability in patients with PDAC and induced longer than expected duration of cancer control [109]. These T cells simultaneously targeted tumor-associated antigens PRAME, SSX2, MAGEA4, NY-ESO-1, and Survivin. Additionally, the development of “off-the-shelf” allogenic CAR-T cells could ameliorate the problem of T-cell exhaustion, as it would no longer rely on the host’s potentially dysfunctional T cells.

### 2.4. Other Immunotherapy Approaches

As previously mentioned, a major challenge in immunotherapy for PDAC is the limited mobility and intratumoral infiltration of CD8+ T cells. In CXCL12-CXCR4 signaling, the chemokine receptor CXCR4, which is stimulated by CXCL12, inhibits the migration of immune cells in preclinical models [110]. In PDAC, a CXCR4 antagonist, AMD3100 (plerixafor), has entered a clinical trial as an adjunct to anti-PD-1/PD-L1 therapy in hopes of augmenting the effects of checkpoint blockade (NCT04177810). Similarly, another CXCR4 antagonist (BL-8040) was evaluated with pembrolizumab and chemotherapy as subsequent-line therapy in metastatic PDAC. In this phase II study, patients who received BL-8040 plus chemoimmunotherapy had an ORR of 32% and median duration of response (mDOR) of 7.8 months, which compared favorably with historical data for second-line therapy [25]. Larger trials are needed to validate these results.

CD40 is a cell surface member of the tumor necrosis factor (TNF) receptor family and, when activated, promotes dendritic cell priming of T cells and macrophages. In a phase Ib study, a CD40 agonist, APX005M (sotigalimab), was used in combination with chemotherapy (GemNab) and nivolumab for metastatic PDAC. Among 24 patients, the ORR was 58%, and median progression-free survival (mPFS) was 11.7 months (95% CI 7.1–17.8 months), which compared favorably with historical mPFS of 5.5 months with chemotherapy only [7,33]. The results are currently being evaluated in a phase II study by the same group [111]. A separate phase I study of APX005M given neoadjuvantly in resectable PDAC showed a significant increase in T-cell-enriched tumors upon resection (82% of tumors were T-cell-enriched) compared to tumors treated with chemoradiation alone (23%, *p* = 0.012) and to untreated tumors (37%, *p* = 0.004) [112]. Results from larger trials are necessary to confirm these promising results.

## 3. Targeted Therapies

To demonstrate the progress of small molecule targeted therapies, we here describe some targets that are of particular excitement to us.

### 3.1. The DNA Damage Repair (DDR) Pathway

Targeting the DDR pathway has become a therapeutic interest for many solid tumors, including PDAC. DNA damage is a common event in normal cells but must be immediately repaired in order to prevent mutation and tumorigenesis. Among other pathways, we highlight two that are of particular interest in double stranded DNA (dsDNA) repair—homologous recombination repair (HRR), which is considered error-proof, and non-homologous end joining (NHEJ), which is more error-prone [113]. Although many genes are involved in HRR, the most well studied ones in PDAC are BRCA1/2 and PALB2, which help form the initial complex at the site of DNA break and activate RAD51 to begin HRR [113]. Any compromise to HRR, such as from BRCA mutations, leads to a state of homologous recombination deficiency (HRD). Retrospective and systematic analyses have found that 15–25% of PDAC have mutations in genes associated with HRD and that there were no significant differences between somatic and germline mutations [114,115,116]. Historically, sensitivity of cancers with HRD to platinum chemotherapy have been well characterized, specifically owing to the inability for HRR-deficient cells to resolve DNA damage induced by platinum therapy [117].

In cancers with HRD, poly(adenosine diphosphate-ribose) polymerase inhibitors (PARPi) have become the breakthrough targeted therapy, particularly in ovarian cancer in which it was first studied clinically [118]. The rationale behind PARPi use in HRD is explained by the concept of synthetic lethality wherein efficient DDR is severely compromised when multiple repair pathways are inhibited (Figure 2) [119]. To simplify this concept, if a mutation in the HRR gene inhibits this pathway, synthetic lethality is established when PARPi directly blocks the base excisions repair (BER) pathway. This scenario creates a reliance on error-prone non-HRR pathways, such as NHEJ (which is also directly promoted by PARPi), and leads to cell death [119,120,121].

PARPi have shown activity in PDAC. The phase III POLO trial demonstrated the sensitivity of PDAC with germline BRCA1/2 mutations to olaparib. Maintenance of olaparib after platinum-based induction therapy showed superior mPFS compared to placebo (7.4 vs. 3.8 months, hazard ratio (HR) 0.53, 95% CI 0.35–0.82, *p* = 0.004), although mOS was similar (18.9 vs. 18.1 months, *p* = 0.68) [122]. Olaparib is FDA-approved in the maintenance setting for patients with metastatic PDAC with germline or somatic BRCA1/2 or PALB2 mutations and is currently being studied in combination therapies. Other PARPi have also entered clinical trials in PDAC, including veliparib, rucaparib, talazoparib, and niraparib [123]

A phase I/II study of the PARPi veliparib plus chemotherapy (fluorouracil and oxaliplatin) was conducted in patients with metastatic PDAC. The ORR was 26% for all patients; however, in patients who were platinum-naïve and had HRD, the response rate was impressively 57% [124]. This further highlighted the importance of patient selection in this personalized approach. A separate phase II trial of veliparib with or without gemcitabine and cisplatin (GEMCIS) in patients with advanced PDAC and germline BRCA/PALB2 mutations unfortunately did not show improved ORR with the PARPi and chemotherapy combination (74.1% vs. 65.2%, *p* = 0.55), nor improved mOS (15.5 months vs. 16.4 months, *p* = 0.6) [125]. However, it suggested increased sensitivity to platinum-based chemotherapy in HRR-deficient PDAC as supported by the higher-than-historical ORR (≈32%) to chemotherapy [6]. In a phase II second-line trial, veliparib did not show benefit when added to fluorouracil and irinotecan (FOLFIRI) [126].

There are several therapeutic challenges in PARPi use. For example, PARPi have notable hematologic toxicity when combined with chemotherapy. A phase I trial of olaparib in combination with irinotecan, cisplatin, and mitomycin was stopped early due to substantial toxicity (grade ≥ 3 neutropenia of 89%), and the addition of veliparib to GEMCIS also caused increased neutropenia (48% vs. 30% in chemotherapy alone) [125,127]. However, no increased toxicity was seen when olaparib was used alone in the POLO trial [122]. Another challenge is to better understand the biomarkers that could predict PARPi sensitivity, as well as refractoriness. The importance of BRCA1/2 and PALB2 in HRR and the benefit of PARPi in BRCA1/2 mutations are well characterized in multiple cancers, including PDAC. However, the role of a number of other genes involved in the HRR (i.e., RAD51, ATM, ATR, CHEK1, ARID1A, etc.) requires better understanding. For example, ATM-deficient tumors appear to be more responsive to radiotherapy, platinum-based chemotherapy, and PARPi [128]. The importance of these genes to HRR likely vary, where ATM deficiencies appear less important than BRCA1/2 in prostate cancer, given the lower response rate to PARPi for ATM-deficient tumors [129].

Studies are currently exploring synergistic therapies to be given with PARPi, such as in combination with ICI. The rationale for this approach is supported by evidence that BRCA1/2-deficient cancers express higher levels of neoantigens, thereby increasing immunogenicity. The DNA damage created by PARPi further generates an interferon response that leads to increased T-cell recruitment and tumor-infiltrating lymphocytes [130]. For example, preclinical studies demonstrate synergy between PARP inhibition and anti-CTLA-4 therapy in BRCA1/2 mutant ovarian cancer [131]. An interaction between PARP inhibitor and tumor-associated immunosuppression likely provides evidence to support the combination of PARP inhibitors and anti-PD-1/PD-L1 combinations. PARPi-related upregulation of PD-L1 expression in breast cancer cell lines and animal models appears to occur by knocking out GSK3β activity, which significantly increases PD-L1 expression and resistance to PARP inhibition. Hence, the blockade of PD-L1 re-sensitized tumor cells to PARP inhibition [132]. A phase II study of olaparib plus pembrolizumab is underway in patients with PDAC and who have HRR gene mutation(s) (NCT04666740).

Aside from PARPi, there are a number of other drugs being developed to target specific elements of HRR, including small-molecule ATR/ATM inhibitors (i.e., M-6620 and BAY-1895344), which have entered early phase clinical studies, and CHK1 inhibitors (i.e., prexasertib) [133]. Data on their efficacies are evolving, and the potential to artificially induce HRD in any cancer with these drugs may further open more doors for PARPi to induce synthetic lethality.

### 3.2. Targeting NTRK

NTRK inhibitors are currently the only other class of targeted agents aside from PARPi that are USFDA-approved for PDAC. NTRK encodes the family of TRK receptors, which bind neurotrophin family ligands and normally promote maintenance and development of the nervous system. TRK receptors activate downstream signaling pathways including MAPK, PI3K, and PCK in order to facilitate neuron growth. The most common aberrant NTRK expression is a gene fusion that causes constitutive activation of TRK proteins and leads to tumor proliferation and survival [134].

Two NTRK inhibitors are USFDA-approved for NTRK-gene-fusion-positive PDAC, loratrectinib and entrectinib, according to phase I/II basket trials. In a pooled analysis of three phase I/II studies of loratrectinib in solid tumors with NTRK gene fusions, the ORR was 79% (121 of 153 patients), and one of two PDAC achieved an objective response. Median DOR was 35.2 months across the entire population [135]. Similarly, in an integrated analysis of three phase I/II trials involving entrectinib in patients with advanced NTRK fusion-positive solid tumors, ORR was 57% (31 of 54 patients), and two of three PDAC achieved an objective response [136]. Although the USFDA approved these inhibitors for use across all NTRK-fusion-positive solid tumors, their actual clinical use is limited by the rarity of these fusions (<1%) in the common cancer types, and only 0.8% in PDAC [137]. Nonetheless, the efficacy of these drugs is impressive for the minority of patients who harbor this genetic aberration.

### 3.3. Targeting KRAS

Activating rat sarcoma vial oncogene (RAS) mutations, including KRAS, are the most commonly mutated oncogenes in all cancers, although they are unevenly distributed among different types of cancers [138]. Specifically, in PDAC, KRAS mutations occur in more than 90% of tumors. There is strong evidence that KRAS is implemented in tumorigenesis and progression [139]. KRAS represents the upstream signaling in the RAS/RAF/MEK/ERK signaling pathway and is normally in a quiescent state but becomes activated by receptors such as EGFR. Activating KRAS mutations result in enhancement of downstream pathways that lead to cell proliferation. These pathways include the RAF-MEK-ERK MAPK pathway, the PI3K-AKT-mTOR pathway, and the Ral guanine nucleotide exchange factor pathway [139]. The most frequent KRAS mutation in PDAC is a point mutation in codon 12, including G12D, G12V, G12R, G12A, and G12C variants [140].

Targeting KRAS has been structurally challenging due to physical characteristics of the KRAS protein, namely, its lack of deep hydrophobic pockets. Initial attempts were made to indirectly target KRAS (i.e., via farnesyl transferase inhibitors, RAF inhibitors, or mTOR inhibitors), but were clinically unsuccessful [139]. In recent years, technological advancements in X-ray crystallography and mass spectrometry enabled identification of a pocket in KRAS G12C where covalent small molecules can bind [141]. This led to the development of KRAS G12C inhibitors including sotorasib, adagrasib, JNJ-74699157, and LY3499446. In non-small cell lung cancer (NSCLC), sotorasib gained accelerated USFDA approval on the basis of a phase II trial demonstrating an impressive ORR of 37.1% and disease control rate (DCR) of 80.6% in pretreated patients with KRAS G12C mutant NSCLC [142]. Adagrasib showed similar efficacy in phase I and II trials for KRAS G12C mutant NSCLC (OR 45%, DCR 96%) [140]. It is currently being examined in phase I and II trials in KRAS G12C solid tumors, including PDAC (NCT03785249); interim results showed that in 10 PDAC patients, there was 50% partial response (PR) and 100% DCR [143].

The potentials for these inhibitors are promising. However, resistance to first-generation KRAS G12C inhibitors have already been identified, which often involves an acquired KRAS Y96D mutation that interferes with drug binding. A novel class of drugs called “tricomplex” inhibitors are engineered to combat this resistance by forming a complex with the mutant KRAS G12C/Y96D and a chaperone protein (cyclophilin A) that is ubiquitous inside cells. RMC-6291 is the first of this drug class, and its efficacy is supported by preclinical data [144]. Nonetheless, there is still a need to develop inhibitors of other spontaneously occurring KRAS mutations such as G12D mutations.

### 3.4. Targeting Downstream Effectors of KRAS

MEK is a downstream effector of KRAS in the RAS/RAF/MEK/ERK pathway. Given the challenges of targeting KRAS directly, there have been several attempts at targeting its downstream effectors, including MEK, where there are readily available potent inhibitors. However, early trials with MEK1/2 inhibitors in metastatic PDAC failed to show convincing benefit. For example, a phase II study of trametinib (MEK1/2 inhibitor) plus gemcitabine did not show significant OS benefit (HR 0.98, *p* = 0.453) [145]. Similarly, another MEK inhibitor, selumetinib, was compared against capecitabine and also showed no benefit in mOS (HR 1.03, *p* = 0.92) [146]. Studies suggest that targeting the RAS pathway gives rise to parallel escape mechanisms from the tumor cells, particularly through autophagy, which helps to explain this resistance. Hence, there are now several trials that combine inhibitors of the RAS pathway (including MEK inhibitors) with an autophagy inhibitor such as hydroxychloroquine. Xavier and colleagues described two cases of trametinib plus hydroxychloroquine in KRAS-mutated chemo-resistant PDAC patients, wherein the patients achieved disease stabilities that were clinically meaningful [147]. A phase II trial is formally investigating this combination [NCT04566133].

### 3.5. Targeting TGFβ Signaling

TGFβ is a signaling molecule that has dual action in cancer, both as a tumor suppressor and a tumor promotor. In its tumor suppressor role, it is a potent regulator of cell cycle arrest in healthy cells and early stage cancer cells. However, its tumor promotor role is of greater interest in research. In mouse models, TGFβ induces epithelial-to-mesenchymal transition, wherein epithelial cells lose their cell-to-cell adhesion properties and become more motile [148]. This transition is key for tumor cell migration and evasion of the immune system. TGFβ also has potent immunosuppressive effects. For example, it promotes immunosuppressive Tregs that repress the function of other effector T cells, such as NK cells [148]. This effect is clinically supported by urothelial cancer samples showing that high levels of TGFβ were associated with decreased response to PD-L1 blockade [149]. In mouse models, it was suggested that blockade of TGFβ augments the effect of anti-PD-L1 therapy [150].

Galunisertib is a small-molecule TGFβ inhibitor that is studied clinically. Unfortunately, a phase Ib study of galunisertib with durvalumab in recurrent/refractory metastatic PDAC demonstrated limited clinical activity (mOS 5.72 months, mPFS 1.87 months, DCR 25%) [17]. Nonetheless, there is still interest in exploiting the TGFβ pathway, such as with newer generation inhibitors (i.e., TGFβ receptor inhibitors, TGFβ checkpoint traps), or in combination with other targeted agents such as anti-VEGF drugs [10].

## 4. Metabolic Pathways

There is renewed interest in targeting cancer cell metabolism based on the general principle that cancer cells rewire many metabolic pathways to promote their own survival and propagation. For example, cancer cells maintain high glycolytic activity, described by the Warburg effect [10]. These metabolic alterations also have a significant impact on the TME, as it competes with other cells, such as T cells, for limited metabolic resources, such as glutamine [10,151]. Furthermore, aberrant KRAS signaling has also been associated with dysregulation of metabolic pathways and leads to increased reliance on metabolites such as glutamine and asparagine [152,153]. Below, we discuss two metabolic pathways in PDAC that have shown some encouraging results.

### 4.1. Targeting Asparagine

Asparagine is a non-essential amino acid that many cancer cells, including in PDAC, are unable to produce in large enough quantities to support their aberrant metabolism [151]. It is made intracellularly from aspartate and glutamine and catalyzed by asparagine synthase, which is heavily expressed in PDAC as an adaptive response to its hypo-vascular TME [153,154]. Eryaspase is an L-asparaginase that is encapsulated in red blood cells and is currently under investigation in PDAC. In a phase II trial of chemotherapy (gemcitabine or fluorouracil plus oxaliplatin) with or without eryaspase for advanced PDAC in the second-line setting, eryaspase plus chemotherapy was well tolerated and showed a survival advantage of 6.0 months vs. 4.4 months (HR 0.60, *p* = 0.0078) [153]. A phase III confirmatory study unfortunately did not meet its primary OS endpoint when chemotherapy was given with or without eryaspase (mOS 7.5 vs. 6.7 months, *p* = 0.375) in the subsequent-line setting, although there was a trend towards improved OS in the group receiving eryaspase with irinotecan-based therapy (i.e., fluorouracil plus irinotecan) [155]. There is currently an ongoing phase I trial of eryaspase plus mFOLFIRINOX (fluorouracil, oxaliplatin, irinotecan) in the front-line setting for PDAC. Interim analysis was encouraging, with 50% PR (5 of 10 patients) and 100% DCR [156]. There were no dose-limiting toxicities.

### 4.2. Targeting Glutamine

Like asparagine, glutamine is another example of an essential amino acid that is in increased demand by tumor cells. As the most abundant amino acid in the blood, it has been well characterized in multiple biological processes that are important for cancer growth and proliferation, such as its role in maintaining redox homeostasis via the glutamine-dependent pathway of cytosolic NADPH production [157,158]. In a proof of principle, Chakrabarti and colleagues demonstrated that by inhibiting glutamine metabolism (via BPTES or CB-839), and thereby reducing NADPH pools, there is increased supra-physiological reactive oxygen species formation, translating to antitumoral activity in vivo and in vitro [158]. Another group demonstrated that ablation of glutamate ammonia ligase, which is required for de novo glutamine synthesis, suppresses the development of KRAS-driven murine PDAC [159]. To date, these results have not yet been replicated in clinical studies.

### 4.3. Targeting Adenosine Generating Enzyme

Adenosine plays an immunosuppressive role in tumorigenesis. CD73 cooperates with CD39 to promote metabolism of proinflammatory ATP to adenosine. Preclinical models have shown increased expression of CD73 in tumor cells, as well as a proficiency in converting ATP to adenosine. Adenosine interacts with G-protein-coupled receptors to promote suppressive immune cells such as MDSCs and Tregs [160,161]. Chen et al. described that a higher CD73 expression was negatively correlated with infiltrating levels of CD8+ T cells in PDAC cell lines [160]. Anti-CD73 and anti-CD39 agents have shown antitumoral activity in preclinical studies, although there is currently a lack of clinical data to support this [162]. There are some phase I trials underway for anti-CD73 agents in combination with existing therapies such as immunotherapies [NCT04148937].

## 5. Conclusions

The aggressive biology of PDAC, its high mortality rate, and the lack of good treatment options has made this cancer a prime focus for the development of newer therapies. In this review, we offered a glimpse into the evolving therapeutic areas in immunotherapy and targeted agents in PDAC.

The success stories of immunotherapies shared by other types of solid tumors are hindered in PDAC due to its unique TME, which is essential to the survival and persistence of the cancer. It exemplifies the properties of a “cold tumor” that has masterfully silenced the immune response within the immune milieu and allow it to evade the effects of immunotherapies. Nonetheless, combinations of ICIs with vaccine therapies, CXCR4 antagonists, or cellular therapies are paving the way to a new generation of immunotherapy approaches that show promise preclinically.

At the same time, evolution of targeted therapies has expanded the repertoire of available molecular targets. For example, PARPi are the just the first class of targeted therapies that have demonstrated clinical activity when specifically studied in PDAC. Yet, there are several new drugs under various stages of development that can inhibit specific genes in the HRR pathway, thereby inducing a state of artificial HRD that could further sensitize cancer cells to PARP inhibition. Additionally, the recent clinical success of KRAS G12C inhibition in NSCLC represents a triumph towards a driver mutation that is frequent yet has been historically difficult to target, showing us the parallel evolution of technology in drug development alongside clinical therapy.

The establishment of mFOLFIRINOX and GemNab as the standard of care for advanced PDAC likely represented the pinnacle of chemotherapy regimens for this disease. The future of PDAC treatment hinges on (1) better understanding of the TME and the tumor immune milieu to allow more effective immunotherapy approaches to overcome the “cold tumor” properties, (2) further characterization of the various signaling and metabolic pathways in PDAC to help uncover new targets and synergies, and (3) leaning on new technologies in drug development (such as X-ray crystallography) and drug delivery (i.e., with novel nanocarriers) [163]. It is essential that we improve the prognosis of this notoriously challenging and deadly cancer.

## Figures and Tables

**Figure 1 cancers-14-02619-f001:**
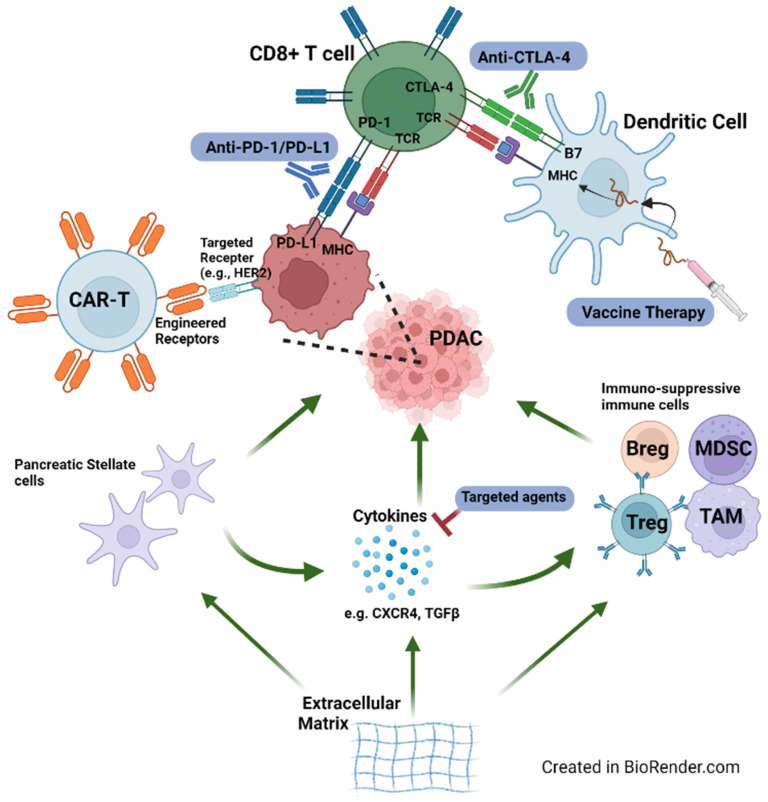
Graphical representation of the immune microenvironment for PDAC. Factors promoting tumoral growth (e.g., pancreatic stellate cells) are indicated with green arrows. Select immunotherapies and targeted therapies are also represented in the figure.

**Figure 2 cancers-14-02619-f002:**
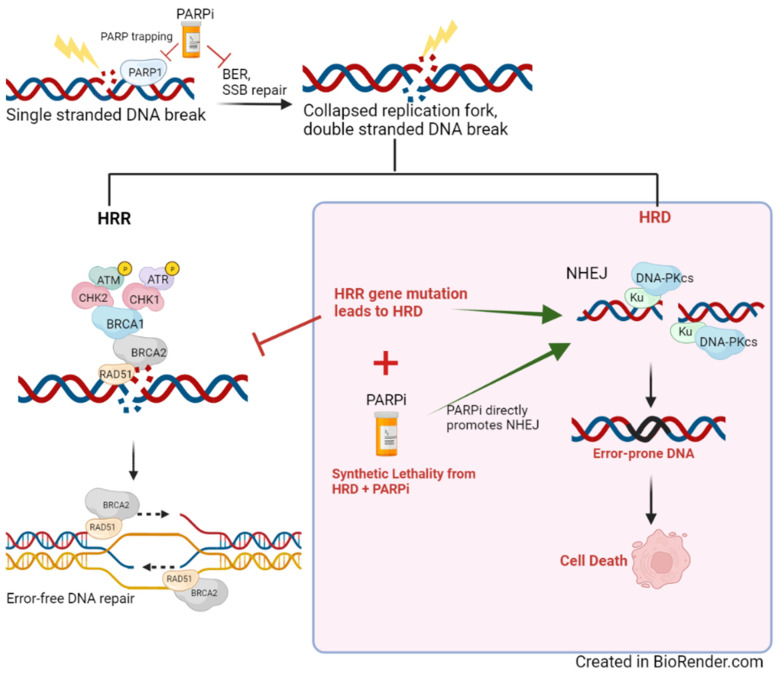
Graphical representation of HRD (homologous repair deficiency), which lends to synthetic lethality with PARPi use. PARPi directly inhibits BER in ssDNA repair, which leads to double-stranded DNA during replication. In the setting of HRD, DNA repair is relegated to error-prone pathways (e.g., NHEJ), which leads to cell death.

**Table 1 cancers-14-02619-t001:** Characteristics and results of published and completed trials with immune checkpoint inhibitors in PDAC.

Treatment	Population	Trial Phase, Year, Author, Ref.	Number of Patients	mPFS (Months)	mOS (Months)	Results
**(1) Durvalumab 1500 mg + galunisertib 50 mg 1×/day**	mPDAC	Phase Ib, 2021, Melisi, [17]	(1) 3	(1) NR	(1) NR	15/32 patient had PD, and DCR was 25.0%.
**(2) Durvalumab 1500 mg + galunisertib 50 mg 2×/day**			(2) 4	(2) NR	(2) NR	
**(3) Durvalumab 1500 mg + galunisertib 80 mg 2×/day**			(3) 3	(3) NR	(3) NR	
**(4) Durvalumab 1500 mg + galunisertib 150 mg 2×/day**			(4) 32	(4) 1.8	(4) 1.8	
**Nivolumab + ipilimumab + radiation**	mPDAC	Phase II, 2021, Parikh, [18]	25	2.5	4.2	DCR was 20% (5/25) of PDAC patients.
**Anti-PD-L1**	Pre-treated LAPC/mPDAC	Phase I, 2012, Brahmer, [19]	14	NR	NR	No objective responses seen in patients with PDAC.
**Pembrolizumab + multiple chemo arms**	Pre-treated mPDAC	Phase 1b, 2017, Weiss, [20]	11	NR	8	No additional data reported for PDAC.
**Pembrolizumab + GemNab**	Pre-treated and untreated mPDAC	Phase Ib-II, 2018, Weiss, [21]	17	9.1	15	DCR was 100% in 11 chemo naïve PDAC patients.
**Nivolumab + mogamulizumab**	Pre-treated mPDAC	Phase I, 2019, Doi, [22]	15	1.8	6.5	DCR was 40% (6/15) and ORR seen in 1/15 patient with PDAC.
**Durvalumab +** **ibrutinib**	Pre-treated LAPC/mPDAC	Phase Ib-II, 2019, Hong, [23]	49	1.7	4.2	ORR seen in 2% of patients with PDAC.
**Durvalumab (D) + tremelimumab (T) or durvalumab (D) monotherapy**	Pre-treated mPDAC	Phase II, 2019, O’Reilly, [24]	65	9.4 (D+T) 3.6 (D)	8.8 (D+T) 6.3 (D)	Combination treatment resulted in an ORR of 3.1%, while monotherapy resulted in an ORR of 0%.
**(1) Anti-CXCR4** **+ pembrolizumab** **(2) Anti-CXCR4** **+ pembrolizumab + chemo**	Pre-treated mPDAC	Phase IIa, 2020, Bockorny, [25]	59	NR	(1) 3.3 (2) 7.2	DCR was 34.5% in patient treated with anti-CXCR4 + Pembrolizumab and 32% in patient with combination of anti-CXCR4 and pembrolizumab with chemotherapy.
**Pembrolizumab**	Pre-treated MSI-H LAPC/mPDAC	Phase II, 2020, Marabelle, [11]	22	2.1	4.0	mDOR was 13.4 months in patients with PDAC.
**Pembrolizumab +** **oncolytic virus (Pelareorep) + chemo**	Pre-treated LAPC/mPDAC	Phase Ib, 2020, Mahalingam, [26]	11	2	3.1	The ORR and DCR were, respectively, 9% and 27%.
**Ipilimumab**	Pre-treated LAPC/mPDAC	Phase Ib, 2010, Royal, [27]	27	NR	NR	No responders to single agent Ipilimumab observed.
**Ipilimumab** **(1) Monotherapy** **(2) + GVAX**	Pre-treated LAPC/mPDAC	Phase Ib, 2013, Le, [28]	30	NR	(1) 3.6 (2) 5.7	3 patients in combination arm had prolonged SD. 2 patients in monotherapy arm had SD
**Tremelimumab + gemcitabine**	chemo naïve mPDAC	Phase Ib, 2014, Aglietta, [29]	34	NR	7.4	2 patients had PR.
**Ipilimumab + gemcitabine**	Previously treated LAPC/mPDAC	Phase Ib, 2015, Mohindra, [30]	13	NR	NR	PR was seen in 2 pts (15%) and stable disease in 5 pts (38%).
**Ipilimumab + gemcitabine**	Pre-treated mPDAC	Phase Ib, 2016, Kaylan, [31]	16	2.5	8.3	The ORR was 14% (3/21), and seven patients had SD.
**Ipilimumab + gemcitabine**	Pre-treated mPDAC	Phase Ib, 2020, Kamath, [32]	21	2.78	6.90	PR seen in 2/16 patients and SD seen 5/16 patients.
**(1) Nivolumab + GemNab + APX005M (0.1 mg/m^2^)**	mPDAC	Phase Ib, 2021, O’Hara, [33]	(1) 6	(1) 10.8	(1) 15.9	ORR 58% (14 patients).
**(2) Nivolumab + GemNab + APX005M (0.3 mg/m^2^)**			(2) 6	(2) 12.4	(2) NR	
**(3) GemNab + APX005M (0.1 mg/m^2^)**			(3) 6	(3) 12.5	(3) 12.7	
**(4) GemNab + APX005M (0.3 mg/m^2^)**			(4) 6	(4) 10.4	(4) 20.1	
**Pegvorhyaluronidase alfa (PEGPH20) + pembrolizumab**	Pre-treated mPDAC	Phase II, 2022, Zhen, [34]	38	1.5	7.2	SD in 2 patients (25%), lasting 2.2 and 9 months.

NR = not reported; GemNab=gemcitabine and nab-paclitaxel; LAPC = locally advanced pancreatic cancer; mPDAC = metastatic pancreatic ductal adenocarcinoma; mPFS = median progression-free survival; mOS = median overall survival; PD = progressive disease; DCR = disease control rate; SD, stable disease; mDOR = median duration of response, ORR = objective response rate; PR = partial response.

**Table 2 cancers-14-02619-t002:** Characteristics of ongoing trials with immune checkpoint inhibitors in PDAC.

Trial Reference	Phase	Treatment	Population	Number of Patients
**NCT04191421** [35]	Ib-II	Spartalizumab + siltuximab	mPDAC	42
**NCT03104439** [36]	II	Nivolumab + ipilimumab + radiation	PDAC	80
**NCT04361162** [37]	II	Ipilimumab + nivolumab + radiation therapy	mPDAC	30
**NCT04477343** [38]	I	SX-682 + nivolumab	mPDAC	20
**NCT04117087** [39]	I	KRAS peptide vaccine + nivolumab + ipilimumab	Resected MMR-p Colorectal cancer and PDAC	30
**NCT04953962** [40]	II	CBP501 + cisplatin + nivolumab	mPDAC	92
**NCT02451982** [41]	II	Arm A: CY/GVAX Arm B: CY/GVAX + nivolumab Arm C: CY/GVAX + nivolumab + urelumab Arm D: BMS-986253 + nivolumab	Surgically resectable PDAC	76
**NCT03970252** [42]	I	Nivolumab, mFOLFIRINOX	Borderline resectable PDAC	36
**NCT03563248** [43]	II	(1) FOLFIRINOX → SBRT → surgery (2) FOLFIRINOX + losartan → SBRT + losartan → surgery (3) FOLFIRINOX + losartan → SBRT + nivolumab + losartan -> surgery (4) FOLFIRINOX → SBRT + nivolumab → surgery	LAPC	160
**NCT04543071** [44]	II	Motixafortide, cemiplimab, gemcitabine, nab-paclitaxel	PDAC	10
**NCT03816358** [45]	I-II	(1) Anetumab ravtansine + nivolumab (2) Anetumab ravtansine + nivolumab + ipilimumab (3) Anetumab ravtansine + nivolumab + gemcitabine	Mesothelin-positive PDAC	74
**NCT03767582** [46]	I-II	Phase I: GVAX + nivolumab + CCR2/CCR5 Phase II: (1) nivolumab + CCR2/CCR5 (2) Nivolumab + GVAX + CCR2/CCR5	LAPC	30

MMR-p = mismatch repair proficient; LAPC= locally advanced pancreatic cancer; mPDAC = metastatic pancreatic ductal adenocarcinoma.

**Table 3 cancers-14-02619-t003:** Characteristics and results of published and completed vaccine trials in PDAC.

Trial Phase, Author, Year, Ref.	Patient Population	Treatment	Vaccine Type; Vaccine Route	Number of Patients	mPFS (Months)	mOS (Months)
**Phase II, Lutz, 2011,** [61]	Resected PDAC	GVAX (+ GM-CSF) + resection + CRT	Whole-tumor-cell; ID	60	17.3	24.8
**Phase II, Le, 2015,** [66]	Pre-treated mPDAC	(1) GVAX (+ GM-CSF) + Cy + CRS-207 (2) GVAX (+ GM-CSF) + Cy	Whole-tumor-cell; ID	90	NR	(1) 6.1 (2) 3.9
**Phase IIb, Le, 2019,** [62]	Pre-treated mPDAC	(1) GVAX (+ GM-CSF) + Cy + CRS-207 (2) GVAX (+ GM-CSF) + Cy (3) Chemo only	Whole-tumor-cell; ID	169	(1) 2.3 (2) 2.1 (3) 2.1	(1) 3.7 (2) 5.4 (3) 4.6
**Phase II, Tsujikawa, 2020,** [67]	Pre-treated mPDAC	(1) GVAX (+ GM-CSF) + Cy + CRS-207 + anti-PD-1 (2) GVAX (+ GM-CSF) + Cy + CRS-207	Whole-tumor-cell; ID	93	(1) 2.2 (2) 2.2	(1) 5.9 (t) (2) 6.1 (t)
**Phase II, Wu, 2020,** [68]	Pre-treated mPDAC	(1) GVAX (+ GM-CSF) + ipilimumab (2) FOLFIRINOX alone	Whole-tumor-cell; ID	82	(1) 2.4 (2) 5.6	(1) 9.4 (2) 14.7
**Phase I, Kaida, 2011,** [69]	Gemcitabine-naïve LAPC/mPDAC	WT-1 vaccine + gemcitabine	Peptide; ID	9	NR	8.2
**Phase I, Nishida, 2014,** [70]	Untreated LAPC/mPDAC and treated recurrent disease	WT-1 vaccine + gemcitabine	Peptide; ID	32	4.2	8.1
**Phase I, Koido, 2014,** [71]	mPDAC: untreated newly diagnosed or recurrence after resection	WT-1 vaccine + gemcitabine	Peptide; ID	10	NR	NR
**NR, Tsukinaga, 2015,** [72]	Untreated mPDAC	WT-1 vaccine + gemcitabine	DC; ID	7	6.8	10.7
**Phase I, Mayanagi, 2015,** [73]	Treatment-naïve LAPC/mPDAC	WT-1 vaccine + gemcitabine	DC; ID	10	NR	8
**Phase I, Yanagisawa, 2018,** [74]	Resected, chemo-naïve PDAC	WT-1 vaccine + chemo	DC; ID	8	NR	NR
**Phase II, Nishida, 2018,** [75]	Untreated LAPC, mPDAC, or recurrence after resection	(1) WT-1 vaccine + gemcitabine (2) Gemcitabine alone	Peptide; ID	85	(1) 5.2 (2) 3.3	(1) 9.6 (2) 8.9
**NR, Hanada, 2020,** [76]	Pre-resected recurrent PDAC	WT-1 vaccine	DC; ID	6	19.9	59
**Phase I-IIa, Nagai, 2020,** [77]	Pre-resected PDAC	WT-1/MUC-1 vaccine + gemcitabine	DC; ID	10	17.7	46.4
**Phase I-II, Asahara, 2013,** [78]	Chemo-refractory, LAPC/mPDAC, or recurrence after resection	(1) KIF20A vaccine (2) No treatment	Peptide; SC	110	(1) 1.8 (2) NR	(1) 4.7 (2) 2.1
**Phase I, Suzuki, 2014,** [79]	Pre-treated LAPC/mPDAC	KIF20A vaccine + gemcitabine	Peptide; SC	9	NR	57
**Phase I, Miyazawa, 2010,** [80]	LAPC/mPDAC	VEGFR2 vaccine + gemcitabine	Peptide; SC	18	3.9	7.7
**Phase II-III, Yamaue, 2015,** [81]	Untreated LAPC/mPDAC	(1) VEGFR2 vaccine + gemcitabine (2) Gemcitabine	Peptide; SC	153	(1) 3.7 (2) 3.8	(1) 8.4 (2) 8.5
**Phase II, Suzuki, 2017,** [82]	Untreated LAPC/mPDAC	KIF20A + VEGFR1/2 vaccine + gemcitabine	Peptide; SC	68	4.7–5.2	9–10
**Phase II, Miyazawa, 2017,** [83]	Pre-resected PDAC	KIF20A + VEGFR1/2 vaccine + gem	Peptide; SC	30	15.8	NR
**NR, Kameshima, 2013,** [84]	LAPC/mPDAC	Survivin vaccine + IFA, IFNα	Peptide; SC	6	NR	NR
**Phase II, Shima, 2019,** [85]	Pre-treated LAPC/mPDAC	(1) Survivin vaccine + IFA, IFNα (2) Survivin vaccine + IFA (3) Placebo only	Peptide; SC	83	(1) 2.2 (3) 2.3	(1) 3.4 (t) (2) 3.2 (t) (3) 3.6 (t)
**Phase I, Rong, 2012,** [86]	Pre-treated LAPC/mPDAC	MUC-1 vaccine	DC; ID	6	NR	NR
**Phase I, Le, 2012,** [87]	Pre-treated PDAC	Mesothelin expressing Lm Vaccine	Lm; IV	9	NR	7
**Phase I, Middleton, 2014,** [63]	Untreated LAPC/mPDAC	(1) Telomerase vaccine (GV1001) sequentially to chemo (2) GV1001 concurrently with chemo (3) Chemo alone	Peptide; ID	1062	(1) 6.4 (2) 4.5 (3) 6.6	(1) 7.9 (2) 6.9 (3) 8.4
**Phase I-II, Wedén, 2011,** [88]	Pre-resected PDAC	KRAS vaccine + GM-CSF	Peptide; ID	23	NR	27.5
**NR, Abou-Alfa, 2011,** [89]	Pre-resected PDAC	KRAS vaccine + GM-CSF	Peptide; ID	24	8.6	20.3
**Phase I, Kubuschok, 2012,** [90]	mPDAC	KRAS vaccine	LCL; SC	7	3.1	4.5
**Phase I-II, Palmer, 2020,** [91]	Pre-resected PDAC	KRAS vaccine + GM-CSF + gemcitabine	Peptide; ID	32	13.9–19.5	33.1–34.2
**Phase Ib, Bassani-** **Sternberg, 2019,** [92]	Pre-resected PDAC	Neoantigens + chemo + anti-PD-1 + aspirin	DC; SC	3	NR	NR
**Phase II, Yanagimoto, 2010,** [93]	Untreated LAPC/mPDAC	Personalized Vaccine + gemcitabine	Peptide; SC	21	7	9
**Phase I, Bauer, 2011,** [94]	Pre-resected recurrent PDAC	Tumor lysate Vaccine + gemcitabine	DC; ID	12	NR	10.5
**NR, Kimura, 2012,** [95]	Chemo-refractory LAPC/mPDAC	Personalized and/or tumor lysate vaccine + chemo + LAK cell therapy	DC; IT	49	NR	11.8
**Phase II, Yutani, 2013,** [96]	Chemo-refractory mPDAC,	Personalized vaccine + chemo	Peptide; SC	41	NR	7.9
**Phase I, Qiu, 2013,** [97]	Pre-treated LAPC/mPDAC	Tumor lysate expressing -Gal + CIK cell therapy	DC; ID	14	NR	24.7
**NR, Lin, 2015,** [98]	Pre-treated stage II PDAC, LAPC, mPDAC	Pancreatic cancer stem cell lysate	Whole-tumor-cell; SC	90	NR	NR
**Phase I, Mehrotra, 2017,** [99]	Pre-treated LAPC/mPDAC	hTERT, CEA, Survivin vaccine + TLR-3 agonist	DC-ID	12	3	7.7
**Phase 1–11, Ota, 2021,** [100]	Advanced or recurrent PDAC	WT1 and/or MUC1 + GEM plus nab-PTX or FOLFIRINOX regimen	Peptide-ID	48	8.1	15.1
**Phase II, Zheng, 2021,** [101]	Pre-resectable PDAC	GVAX + Cy	Whole tumor cell-ID	(1) 29 (2) 28 (3) 30	(1) NR (2) NR (3) NR	(1) 34.2 (2) 15.4 (3) 16.5

NR = not reported; R=retrospective; ID = intradermal; IV = intravenous; IM = intramuscular; IT = intratumoral; SC = subcutaneous; Gal = alpha-galectin; TLR = Toll-like receptor; Cy = cyclophosphamide; CRS-207 = mesothelin-expressing Lm vaccine; Lm = Listeria monocytogenes; DC = dendritic cell, LAK = lymphokine-activated killer; CIK = cytokine-induced killer; chemo = chemotherapy; BSC = best supportive care; IFA = incomplete Freund’s adjuvant = IFNα, interferon-alpha; LAPC = locally advanced pancreatic cancer; mPDAC = metastatic pancreatic ductal adenocarcinoma; mPFS = median progression-free survival; mOS = median overall survival.

**Table 4 cancers-14-02619-t004:** Characteristics of ongoing vaccine trials in PDAC.

Trial Reference	Phase	Treatment	Population	Number of Patients
**NCT03956056** [102]	I	Neoantigen peptide vaccine	Pre-resected PDAC	15
**NCT04117087** [39]	I	KRAS peptide vaccine, nivolumab, and ipilimumab	Pre-resected PDAC	30
**NCT01088789** [103]	II	Multiple cohorts and arms involving allogenic pancreatic tumor cell vaccine transfected with GM-CSF, in combination with cyclophosphamide	Pre-resected PDAC	72
**NCT03592888** [104]	I	mDC3/8-KRAS vaccine	Pre-resected PDAC	12
**NCT02600949** [105]	I	Multiple cohorts testing personalized vaccine + imiquimod with pembrolizumab and APX005M	Advanced PDAC or colorectal cancer	150
**NCT03006302** [106]	II	Epacadostat + pembrolizumab + CY + GVAX + CRS-207	mPDAC	44
**NCT04157127** [107]	I	Autologous DC vaccine	PDAC	43
**NCT02451982** [41]	I	Arm A: CY/GVAX alone Arm B: CY/GVAX + nivolumab Arm C: CY/GVAX + nivolumab + urelumab Arm D: BMS-986253 + nivolumab	Resectable adenocarcinoma of the pancreas	76
**NCT03767582** [46]	I-II	Phase I: GVAX/Nivolumab/CCR2/CCR5 dual antagonist Phase II: Arm A: nivolumab/CCR2/CCR5 dual antagonist Arm B: nivolumab/GVAX/CCR2/CCR5 dual antagonist	Locally PDAC	30

IL = interleukin; cy = cyclophosphamide.

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
