# Peer review of "The Next Frontier in Pancreatic Cancer: Targeting the Tumor Immune Milieu and Molecular Pathways"

_cancers, 2022, doi:10.3390/cancers14112619_

Round 1

Reviewer 1 Report

This review article by Yin et al. introduces the deadly nature of pancreatic cancer, followed by a brief review of the old immunotherapies and targeted therapies in PDAC. The article then describes the new strategies, such as combining immune checkpoint inhibitors with vaccine therapy and chemokine receptor antagonist. Overall, it is a well-written review article describing the evolving therapeutic areas in immunotherapy and targeted agents in PDAC.

Author Response

Thank you for your compliments!

Reviewer 2 Report

In present review, authors discusses the utility of  synthetic lethality and targeting metabolic modulators such as KRAS inhibitors in combination with immunotherapy. I have several reservations, my comments are appended as below:

  1. Author’s should share quantitative data on immunotherapy efficacy.
  2. PDAC and immunotherapy/vaccine trials: authors should present in table with relevant details as no of patients, statistical inference, combination used etc. (completed and ongoing trails).
  3. T cell exhaustion- there are other markers than PD1, CTLA known to have important role in T cell exhaustion. Authors should see if reported in literature.
  4. Inhibitors: share the details on FDA approval. For instance, line 217. Author’s should include all mentioned in manuscript in table.
  5. For immunotherapy, other cofounders as BMI, smoking are known to play an important role. Authors may refer PMID: 33076303 and discuss.
  6. Targeting KRAS, TGFB and metabolic pathways- should include figure.
  7. Targeting metabolism- should discuss notable pathways as glutamine/lactic acid
  8. There should be ‘future directions’ section.

Author Response

Dear reviewer, thank you for taking the time to evaluate our review article and for giving us these thoughtful suggestions, we will address each comment accordingly.

1) We have added additional quantitative data on immunotherapy efficacy, including trials with durvalumab, sintilimab, and CD40 agonist.

2) We have added tables listing completed and ongoing trials with ICI and vaccine therapies.

3) The topic of T-cell exhaustion is more complex than we hope to elucidate in the scope of our paper. However, we have inserted some additional markers such as TIM-3, and LAG3.

4) We have made additional tables as suggested in #2. Regarding FDA approvals, there are only 4 non-chemotherapy drugs approved at the moment (pembrolizumab, loratrectinib, entrectinib, and olaparib), therefore we felt it sufficient to describe in the body of the paper.

5) Thank you for pointing out confounders to immunotherapy. We have looked further into these confounders. However, the point that we hope to make regarding immunotherapy is that PDAC seems to behave a little more differently from other malignancies in its responsiveness to immunotherapy. The referenced article does not address how such confounders might be different in PDAC patients compared to other malignancies that could explain why PDAC patients do not respond as well to immunotherapy.

6) We have included a graphical abstract (seems to have been cut out in initial rendered manuscript) that includes targetable pathways.

7) We have added an additional paragraph on targeting glutamine metabolism.

8) We have incorporated an outline of future directions into our conclusions in the revised draft

Reviewer 3 Report

Overall the review article is straightforward and within the scope of  MDPI-Cancers.

However, I have some queries which should be addressed before publishing this review article:

  • Please, include the latest pancreatic cancer statistics from 2022.
  • IRAK4 inhibitor in pancreatic cancer?
  • Though the topic of this review is Targeting the Tumor Immune Milieu and Molecular Pathways, the authors don’t mention the role of cancer stem cells and their role in immune evasion and what kind of immunotherapies and combination therapies to target them as CSCs are the principal cause for relapse.
  • The authors should also discuss Aldehyde Dehydrogenase and its role in pancreatic cancers.
  • The authors should discuss nanotherapeutics mechanisms of immune induction by nanocarriers as well as nanocarriers in PDAC immunotherapy.

Author Response

Dear reviewer, thank you for taking the time to evaluate our review article and for giving us these thoughtful suggestions, we will address each comment accordingly.

1) We have updated the cancer statistics to reflect 2022.

2) Thank you for pointing out IRAK4 inhibitor. While this is a potentially useful marker for immunotherapy resistance, we did not find much data on use in PDAC. Our overview of immunotherapies is not meant to be exhaustive. We will keep an eye out for additional data on IRAK4 inhibitors in the future!

3) We will briefly address CSC in our revision.

4) We briefly included ALDH in our mention about CSC.

5) Nanocarriers and other new engineered technologies are exciting! But it is a little outside of our focus (more so on molecular targets and pathways). However, we will mention this in our conclusions as part of future directions.

Round 2

Reviewer 2 Report

Accept

Reviewer 3 Report

The authors have addressed all the concerns raised and I am fine with publishing this paper